# Age- and Severity-Related In-Hospital Mortality Trends and Risks of Severe Traumatic Brain Injury in Japan: A Nationwide 10-Year Retrospective Study

**DOI:** 10.3390/jcm10051072

**Published:** 2021-03-04

**Authors:** Chiaki Toida, Takashi Muguruma, Masayasu Gakumazawa, Mafumi Shinohara, Takeru Abe, Ichiro Takeuchi, Naoto Morimura

**Affiliations:** 1Department of Disaster Medical Management, The University of Tokyo, 7-3-1 Hongo, Bunkyo-ku, Tokyo 113-8655, Japan; molimula-tky@umin.ac.jp; 2Department of Emergency Medicine, Yokohama City University Graduate School of Medicine, 4-57 Urafunecho, Minami-ku, Yokohama 232-0024, Japan; mgrmtks@gmail.com (T.M.); gakumazawa-tuk@umin.ac.jp (M.G.); shinoharamafumi@yahoo.co.jp (M.S.); abet@yokohama-cu.ac.jp (T.A.); takeqq@yokohama-cu.ac.jp (I.T.)

**Keywords:** pediatric patient, severely injured patients, in-hospital mortality, traumatic brain injury, injury prevention

## Abstract

Traumatic brain injury (TBI) is the major cause of mortality and morbidity in severely-injured patients worldwide. This retrospective nationwide study aimed to evaluate the age- and severity-related in-hospital mortality trends and mortality risks of patients with severe TBI from 2009 to 2018 to establish effective injury prevention measures. We retrieved information from the Japan Trauma Data Bank dataset between 2009 and 2018. The inclusion criteria for this study were patients with severe TBI defined as those with an Injury Severity Score ≥ 16 and TBI. In total, 31,953 patients with severe TBI (32.6%) were included. There were significant age-related differences in characteristics, mortality trend, and mortality risk in patients with severe TBI. The in-hospital mortality trend of all patients with severe TBI significantly decreased but did not improve for patients aged ≤ 5 years and with a Glasgow Coma Scale (GCS) score between 3 and 8. Severe TBI, age ≥ 65 years, fall from height, GCS score 3–8, and urgent blood transfusion need were associated with a higher mortality risk, and mortality risk did not decrease after 2013. Physicians should consider specific strategies when treating patients with any of these risk factors to reduce severe TBI mortality.

## 1. Introduction

Traumatic brain injury (TBI) is the major cause of mortality and morbidity in severely injured patients worldwide [1,2,3]. It was suggested that multiple risk factors, such as age, injury mechanism, and physiological status on hospital arrival, among others, were associated with the mortality and morbidity of patients with severe TBI [4,5,6,7]. A previous study reported that the mortality rates ranged from 9 to 28.1 per 100,000 people per year in European countries according to the social conditions and health of the population [1]. Furthermore, several studies found that younger and elderly patients with severe TBI showed higher mortality than patients in other age groups [8,9]. In summary, the trend of in-hospital mortality in patients with severe TBI has also been reported to vary by country, injury mechanism, and age-related injury characteristics [2,3,10,11,12]. Therefore, injury surveillance is essential for identifying age- and severity-related injury characteristics and monitoring in-hospital mortality trends over time, as well as to assess risk factors related to primary and secondary injury prevention, to estimate the effectiveness of injury prevention measures [13,14].

There is no other country in the world where the decreasing birthrate and aging of the population has advanced more rapidly than in Japan. However, it remains unclear whether the epidemiological characteristics of severe TBI mortality and mortality risk of severe TBI in Japan is because of the lack of data on the epidemiology and mortality of patients with severe TBI. Therefore, studies that use the Japanese nationwide trauma registry may be useful to other developed countries that analyze age- and severity-related characteristics and mortality risk in severe TBI. Thus, this study aimed to evaluate age- and severity-related in-hospital mortality trends and risk factors associated with mortality in patients with severe TBI using nationwide data comprising a 10-year period between 2009 and 2018. We aimed to address the aforementioned research gaps as well as to provide information that could help developing and evaluating effective injury prevention measures and specific therapeutic strategies.

## 2. Materials and Methods

### 2.1. Study Setting and Population

This retrospective observational study was conducted using data obtained from the Japan Trauma Data Bank (JTDB), which registers data on patients with injury and/or burns and records prehospitalization and hospital-related information. The JTDB data included demographics, comorbidities, injury types, mechanism of injury, means of transportation, vital signs, Abbreviated Injury Scale (AIS) score, prehospital/in-hospital procedures, injury diagnosis as indicated by the AIS, and clinical outcomes. In most cases, physicians trained in AIS coding undertake the online registration of individual patient data. The JTDB data collection started from 55 hospitals in 2003. The number of participating hospitals increased year by year and included up to a total of 280 hospitals in all 47 prefectures in Japan, including 92% of Japanese government-approved tertiary emergency medical centers in March 2019. The Japan Association for the Surgery of Trauma permits open access and updating of existing medical information, and the Japan Association for Acute Medicine evaluates the submitted data [12].

In this study, we used a JTDB dataset that included information from 1 January 2009, to 31 December 2018, which initially yielded the data of 313,643 patients. The inclusion criteria for this study were patients with severe TBI, which was defined as those patients with an Injury Severity Score (ISS) ≥ 16 and TBI. Patients with burns, injury regions with AIS score ≥ 3 other than head injury, or with missing key data were excluded from this study. Figure 1 presents a flow diagram of the patients’ disposition in this study.

### 2.2. Data Collection

We collected the information from the JTDB as follows: demographic data (age (years), sex, year of hospital admission, transportation method, injury mechanism), clinical parameters (Glasgow Coma Scale (GCS) score, ISS [15], AIS score of the injured region, Revised trauma score (RTS) that consisted of the following: age, systolic blood pressure, and respiratory rate [16], survival probability, the requirement of computed tomography, urgent blood transfusion within 24 h from hospital arrival, urgent transcatheter arterial embolization (TAE), urgent surgery; craniotomy and/or craterization, hospital location, length of hospital stay, disposition at discharge), and outcomes (in-hospital mortality, standardized mortality ratio (SMR)). The outcome measures were in-hospital mortality trends and risk factors for in-hospital mortality. The survival probability and predicted mortality were calculated using the Trauma and Injury Severity Score (TRISS) [17], and SMR was calculated by dividing the in-hospital mortality by the mean predicted mortality.

### 2.3. Statistical Analyses

This study estimated the following: (1) patients’ age-related characteristics and outcomes employed during the 10-year study period, (2) 10-year in-hospital mortality trends, and (3) risk factors associated with in-hospital mortality over 10 years. In the primary analysis, which was conducted to identify the characteristics of patients with TBI during the study period, the Mann–Whitney U test and the Kruskal–Wallis test were used for the analyses of continuous variables, whereas a chi-square test was used for categorical variables. In the secondary analysis, which was conducted to identify the in-hospital mortality trends during the study period, the Cochran–Armitage test was used to test for trends in in-hospital mortality by study-year, which was treated as an ordinal variable. In addition, the Cochran–Armitage test was repeated for patients with severe TBI by the age and GCS score categories. In the third analysis, the following variables were applied to the multivariate logistic regression analyses: age groups, year of hospital admission, transportation method, injury mechanism, GCS score at hospital arrival, probability of survival, and requirement of urgent treatment. In addition, the multivariate logistic regression analyses were repeated for severe TBI with most severity following a GCS score of 3–8. The dependent variable in the multivariate logistic regression was in-hospital mortality. The results of these comparisons are expressed as the medians and interquartile ranges (IQRs; 25th–75th percentile) for continuous variables and as patient numbers and percentages for categorical variables. All statistical analyses were performed using STATA/SE software, version 16.1 (StataCorp; College Station, TX, USA). A two-tailed *p*-value < 0.05 indicates statistical significance.

### 2.4. Ethics Statement

This study was approved by the institutional ethics committees of Yokohama City University Medical Centre (approval no. B170900003). The approving authority for data access was the Japanese Association for the Surgery of Trauma (Trauma Registry Committee). The requirement for informed consent from the patients was waived due to the observational nature of the study design.

## 3. Results

During the 10-year study period, 31,953 patients with severe TBI (32.6%) among the total number of severely injured patients with an ISS ≥ 16 (Figure 1) were included. These patients were categorized into the following age groups: neonates/infants; 0, preschool children; 1–5, schoolchildren; 6–12, adolescents; 13–17, adults; 18–64, and older adults ≥ 65 years. There was a significant difference between the age-stratified subgroups in the incidence rate of severe TBI among severely injured patients (93.5%, 56.5%, 50.3%, 32.5%, 23.1%, and 40.7%, respectively, *p* < 0.001). The study cohort included male patients (*n* = 21,589; 68%), patients who were transported from another hospital (*n* = 5028; 16%), patients with blunt trauma (*n* = 31,881; 99.8%), and patients with GCS scores between 14–15/9–12/3–8 (*n* = 15,320/7700/8933; 48/24/28%). The overall median survival probability, actual in-hospital mortality, and SMR were 97%, 17.3%, and 0.82, respectively (Table 1).

There were significant differences in the proportion of male patients (*p* < 0.001), transportation method (*p* < 0.001), injury mechanism (*p* < 0.001), RTS score (*p* < 0.001), survival probability (*p* < 0.001), and actual in-hospital mortality (*p* < 0.001) by age groups.

Table 2 shows that there are significant differences in the proportion of patients who underwent examination and treatment, hospital location, length of hospital stay, and discharge disposition in the age-stratified subgroups.

Figure 2 shows the 10-year in-hospital mortality trend for each age group. In all patients with severe TBI, the in-hospital mortality significantly decreased over the 10-year period in the Cochran–Armitage test (from 21.5% in 2009 to 14.9% in 2018, *p* < 0.001). In the age-related groups, the in-hospital mortality in schoolchildren, adolescents, adults, and older adults significantly decreased over the 10-year period (from 6.8% to 0%, *p* = 0.012, from 6.3% to 1.4%, *p* < 0.001, from 19.6% to 11.2%, *p* < 0.001, and from 26.0% to 17.7%, *p* < 0.001, respectively). There was no significant change in the in-hospital mortality trend among those under 5 years of age.

Figure 3 shows 10-year in-hospital mortality trends according to severity: GCS scores were between 14 and 15, 9 and 12, and 3 and 8. Although decreases were observed in the in-hospital mortality of patients with GCS scores of 14–15 and 9–12 (from 4.1% in 2009 to 2.5% in 2018, *p* < 0.001 and from 15.0% to 9.2%, *p* = 0.002, respectively), there were no significant changes in the in-hospital mortality of patients with a GCS score of 3–8 (from 47.5% to 47.0%, *p* = 0.176).

Table 3 shows the multivariate logistic regression analyses. In all patients with severe TBI, the mortality risk of patients aged <65 years was significantly lower than that of older adults (comparative controls). When the year 2018 was used as a comparative control, patients admitted before 2013 had a higher mortality risk. Patients who underwent an inter-hospital transfer were significantly associated with lower odds of in-hospital mortality (*p* = 0.048, odds ratio (OR) = 0.81, 95% confidence interval (CI) = 0.66–0.99). Regarding severity, the mortality risk of patients with GCS scores of 14–15 and 9–13 was significantly lower than that of patients with a GCS score of 3–8 (*p* < 0.001, OR = 0.19, 95% CI = 0.16–0.22; *p* < 0.001, OR = 0.51, 95% CI = 0.45–0.57).

Moreover, in the overall patients and patients with a GCS score of 3–8, traffic accidents by motor vehicles or bikes were significantly associated with lower mortality risk and fall from a height was significantly associated with a higher mortality risk. Patients with higher survival probability were associated with lower mortality risk. Patients who underwent urgent blood transfusion had a significantly higher mortality risk. In contrast, patients who underwent TAE and surgery were associated with lower mortality risk.

## 4. Discussion

To the best of our knowledge, this is the first nationwide study to evaluate the age- and severity-related in-hospital mortality trends and mortality risks of patients with severe TBI during the 10-year study period in an aging society like Japan. The four findings obtained in this study are summarized as follows: (1) patients with severe TBI had significant age-related differences in characteristics, mortality trends, and mortality risks; (2) in-hospital mortality trends of all patients with severe TBI significantly decreased from 2009 to 2018; however, the in-hospital mortality trend of patients aged less than 5 years and with a GCS score between 3 and 8 did not; (3) patients older than 65 years old, those who fell from height, with a GCS score between 3 and 8, and who needed urgent blood transfusion had a higher mortality risk; (4) among patients with a GCS score between 3 and 8 (the most severe cases), those younger than 5 years old or older than 65 years old, those who fell from height, and those who underwent urgent blood transfusion also had a higher mortality risk, and this risk did not decrease significantly after 2013.

Previous studies reported the age- and severity-related characteristics and mortality in different countries [1,2,10,11,18,19]. However, these data indicated that there is a high mortality risk and age-related characteristics and outcome in patients with severe TBI, and it is important to take preventive measures to decrease the mortality of patients with severe TBI according to data from a high-quality injury surveillance system in different countries and regions [12]. Previous studies have reported in-hospital mortality of patients with severe TBI ranging from 17.0% to 55% [18,19]. While different survey methods and severe TBI definitions make direct comparisons difficult, we observed in-hospital mortality rates of 46.4% and 49.6% in all patients with severe TBI and those with a GCS score between 3 and 8, respectively, which was not considerably lower compared with that of other developed countries [17,18]. The mortality trend of patients with severe TBI that are younger than 5 years old and have a GCS score between 3 and 8 did not improve during the 10-year study period; therefore, it may be useful to consider the risk factors of these patients to decrease the mortality of this population in Japan. Ensuring that patients with a low GCS score—especially those that are also younger than 5 years old or older than 65 years old—receive specific and appropriate medical treatment considering their risk factors may effectively decrease the mortality rate of patients with severe TBI. Regarding why young patients with a low GCS score had a higher mortality risk, it is suspected that it could due to pediatric abusive head trauma (AHT), such as shaken baby syndrome that was observed in severe TBI patients aged ≤ 5 years in this study. One study reported that the mortality of patients with AHT admitted to the Pediatric Intensive Care Unit was 24%; however, AHT is a potentially preventable injury [20]. Moreover, a previous study showed that mortality in severe TBI patients aged ≥ 65 years was 71–87%, and the mortality increased age-dependently [9]. Therefore, aggressive therapy and injury prevention may be more effective for these patients compared with other age groups.

Regarding the injury mechanism, this study showed that traffic accidents caused by motor vehicles or bikes were associated with lower mortality risk and fall from height was associated with a higher risk. The main cause of severe TBI mortality differs based on the location that the studies were conducted [1,2,10,11]. In many European countries, falls are the major cause, whereas in the United States, firearm-related wounds are the leading cause. In China, traffic accidents caused by motor vehicles were the most common cause of TBI mortality. In Japan, the establishment of laws requiring seat belt usage and strengthened penalties for drunk driving, and improvement of motor vehicle engineering might affect in-hospital mortality. Although traffic accidents involving motor vehicles or bikes were significantly associated with a lower mortality risk, we observed a high frequency of traffic accidents involving bicycle riders and pedestrians. Thus, traffic policies focused on these modes of transportation may positively affect mortality rates. Furthermore, prevention measures, including structural control of building and prevention programs for occupational injury and suicide from a tall building, might be effective for decreasing TBI mortality. Furthermore, there were age-related differences in the injury mechanism of severe TBI in this study. For example, major causes of severe TBI in patients ≤ 5 years and ≥18 years were a fall from the stairs or tumbling, and in contrast, the major causes of severe TBI in schoolchildren and adolescents were traffic accidents by bicycles. Our findings indicate that injury prevention in every age group is essential.

In this study, a low GCS score and urgent blood transfusion were associated with a higher mortality risk of TBI. In contrast, inter-hospital transport and urgent treatment were associated with lower mortality risk. This finding suggests that patients with severe TBI who had unstable vital signs and those who could not undergo urgent therapy may die. Many studies have shown that a GCS score ≤ 8, hypotension, or coagulopathy were the prognostic factors of mortality after TBI, and appropriate urgent operative management and neurosurgical unit care significantly reduced mortality and neurological deficits after severe TBI [4,5,9]. In contrast, only 9% of patients with severe TBI who were admitted to a general hospital with trauma facilities but without a neurosurgical care unit underwent inter-hospital transfer and 23% of TBI were managed in non-specialist critical care [21]. In Japan, most patients with severe TBI are transported by ambulance to the nearest emergency hospital without neurosurgeons and/or neurosurgical care units for initial stabilization. Whether a direct transfer from the scene to a neurosurgical facility or secondary transfer from a general hospital to a neurosurgical facility is the most appropriate course of action remains unclear. Moreover, we observed that the proportion of inter-hospital transfer differed according to age group; thus, a future analysis must assess the effect of different transfers on mortality. Previous studies indicated the limitations of using the GCS as a predictor of neurological outcomes for diagnosis, surgical management, and posterior prognosis, especially in younger patients and those with hypoxic injury [22,23]. Although we applied the GCS universally in this study, future studies may have to normalize the results in relation to other scales, such as the modified GCS for young children or the full outline of unresponsiveness scale. However, patients with severe TBI who need urgent blood transfusion and surgical intervention could show better outcomes if appropriate decision-making is done and they receive timely neurosurgical care as soon as possible.

This study had several limitations. The retrospective study design and certain missing data in the JTDB impaired the precision of this analysis. Therefore, there was a selection bias because not all Japanese hospitals that treat trauma participate in the JTDB. Our results indicate that the aging of the Japanese population probably affected the proportion of older patients with severe TBI, which increased from 47% in 2009 to 64% in 2018. A previous report showed that patients aged ≥ 75 years had the highest TBI-related mortality among all age groups [23,24]. Thus, future studies may consider conducting sub-analyses within the aged group. Moreover, the ISS score system used to assess injury severity may have some limitations, especially related to mortality probability or scoring system validity [25,26]. For a high-quality injury surveillance system that can accurately evaluate quality indicators of injury care, such as the structure of the care system, care process, and injury outcomes, a dataset registering all inpatients in the survey target area is essential.

There is a need for improved injury surveillance systems, with comprehensive datasets including all Japanese injured patients and minimum missing data, for the establishment of injury prevention strategies and the evaluation of the quality of injury care and related outcomes in the future. In addition, we did not conduct subclass analyses based on factors such as the proportion of preventable trauma deaths or optimal treatment affecting mortality. Previous studies failed to prove that the outcomes of patients improved with the use of simple risk factor assessments [27,28]. Accordingly, although this retrospective study assessed the mortality risk of patients with severe TBI, our results regarding mortality risk cannot be used to directly guide clinicians toward optimal TBI prevention and treatment strategies. Moreover, different neurosurgical pathologies—such as acute epidural and subdural hematomas, chronic subdural hematoma, and intracranial hemorrhage—have distinct relevance depending on specific demographic factors and baseline medical conditions. Therefore, in the future, we should analyze the specific mortality risk of patients with severe TBI considering other unaccounted extra-neurological mechanisms, such as coagulopathies or ionic problems, as well as appropriate strategies to reduce severe TBI mortality of at-risk patients. Finally, we could not assess the differences in mortality owing to intra-hospital complications and medical therapeutic strategies among the facilities.

## 5. Conclusions

In this nationwide study during the 10-year study period, there were significant age-related differences in characteristics, mortality trend, and mortality risk of patients with severe TBI. The in-hospital mortality trend of all patients with severe TBI significantly decreased from 2009 to 2018; however, the in-hospital mortality trend of patients aged less than 5 years and with a GCS score between 3 and 8 did not improve. Moreover, in all patients with severe TBI, age ≥ 65 years, fall from a height, a GCS score between 3 and 8, and urgent blood transfusion was associated with a higher mortality risk, and the mortality risk did not decrease after 2013. Patients with severe TBI, a GCS score between 3 and 8, and those who need an urgent blood transfusion and surgical intervention might have better outcomes if they receive specific treatment strategies. Future nationwide studies with subclass analyses should be conducted to provide information for developing these strategies as well as prevention measures to improve the outcomes of patients with severe TBI.

## Figures and Tables

**Figure 1 jcm-10-01072-f001:**
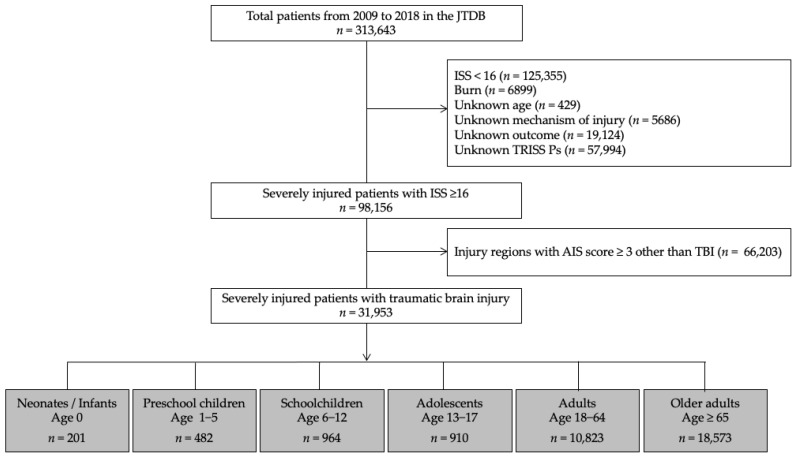
Flow diagram of the disposition of study patients. JTDB, Japanese Trauma Data Bank; ISS, Injury Severity Score; AIS, Abbreviated Injury Scale; TBI, Traumatic brain injury; TRISS, Trauma and Injury Severity Score; Ps, Probability of survival.

**Figure 2 jcm-10-01072-f002:**
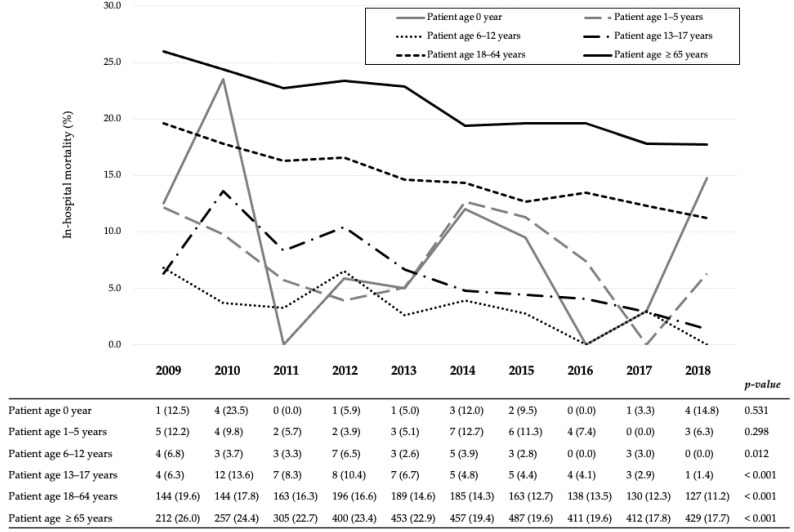
Yearly in-hospital mortality trends according to the age group of patients with traumatic brain injury. All variables are expressed as the number of deaths (mortality, %).

**Figure 3 jcm-10-01072-f003:**
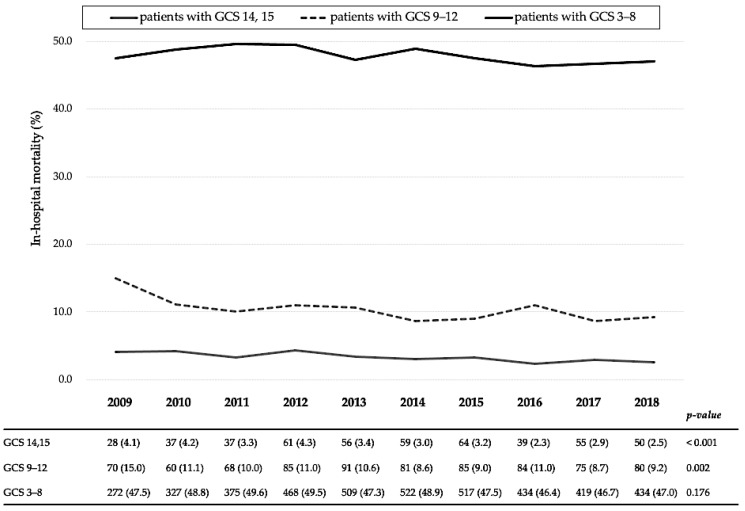
Yearly in-hospital mortality trends according to the Glasgow Coma Scale score categories of patients with traumatic brain injury. GCS, Glasgow Coma Scale. All variables are expressed as the number of deaths (mortality, %).

**Table 1 jcm-10-01072-t001:** Demographics and other data of patients with traumatic brain injury by age groups.

Variables	Overall *n* = 31,953	Neonates/Infants *n* = 201	Preschool Children *n* = 482	Schoolchildren *n* = 964	Adolescents *n* = 910	Adults *n* = 10,823	Older Adults *n* = 18,573	*p*-Value
**Age, years**	69 (49–80)	0	3 (2–4)	9 (7–11)	16 (14–17)	49 (35–59)	78 (72–84)	<0.001
**Male**	21,589 (68)	149 (74)	304 (63)	653 (68)	648 (71)	8440 (78)	11,395 (61)	<0.001
**Year of hospital admission**								
2009	1725 (5)	8 (4)	41 (9)	59 (6)	64 (7)	736 (7)	817 (4)	<0.001
2010	2089 (7)	17 (8)	41 (9)	81 (8)	88 (10)	809 (7)	1053 (6)	<0.001
2011	2571 (8)	17 (8)	35 (7)	90 (9)	84 (9)	1002 (9)	1343 (7)	<0.001
2012	3146 (10)	17 (8)	52 (11)	107 (11)	77 (8)	1183 (11)	1710 (9)	<0.001
2013	3572 (11)	20 (10)	59 (12)	115 (12)	105 (12)	1292 (12)	1981 (11)	0.026
2014	3962 (12)	25 (12)	55 (11)	130 (13)	106 (12)	1295 (12)	2351 (13)	0.419
2015	4060 (13)	21 (11)	53 (11)	106 (11)	114 (13)	1284 (12)	2482 (13)	0.002
2016	3394 (11)	19 (9)	54 (11)	99 (10)	98 (11)	1025 (9)	2099 (11)	<0.001
2017	3648 (11)	30 (15)	44 (9)	101 (10)	102 (11)	1060 (10)	2311 (12)	<0.001
2018	3786 (12)	27 (13)	48 (10)	76 (8)	72 (8)	1137 (11)	2426 (13)	<0.001
**Transportation**								
Transportation from the scene	24,708 (77)	96 (48)	288 (60)	691 (72)	784 (86)	9164 (84)	13,685 (74)	<0.001
Transportation from another hospital	5028 (16)	76 (38)	111 (23)	183 (19)	75 (8)	1167 (11)	3416 (18)	<0.001
**Mechanism of injury**								
Blunt	31,881 (99.8)	201 (100)	480 (99.6)	963 (99.9)	909 (99.9)	10,782 (99.6)	18,546 (99.9)	0.002
**Injury mechanism of blunt trauma**								
Traffic accident								
Motor vehicle	1400 (4)	21 (11)	40 (8)	38 (4)	33 (4)	721 (7)	547 (3)	<0.001
Bike	2047 (6)	0	2 (0.4)	3 (0.3)	196 (22)	1330 (12)	516 (3)	<0.001
Bicycle	3846 (12)	2 (1)	23 (5)	308 (32)	373 (41)	1461 (14)	1679 (9)	<0.001
Pedestrian	3154 (10)	1 (0.5)	64 (13)	243 (25)	75 (8)	1111 (10)	1660 (10)	<0.001
Fall from a height	2067 (6)	19 (9)	114 (14)	89 (9)	40 (4)	973 (9)	832 (4)	<0.001
Fall from stairs	5381 (17)	56 (28)	128 (27)	105 (11)	23 (3)	1801 (17)	3268 (18)	<0.001
Fall at the same level	11,087 (34)	62 (31)	77 (16)	92 (10)	35 (4)	2022 (17)	8799 (47)	<0.001
Others	2971 (9)	40 (20)	34 (7)	86 (9)	135 (15)	1404 (13)	1272 (7)	<0.001
**Glasgow Coma Scale score on hospital arrival**								
14, 15	15,320 (48)	99 (49)	244 (51)	546 (57)	419 (46)	4755 (44)	9257 (50)	<0.001
9–12	7700 (24)	51 (25)	112 (23)	245 (25)	243 (27)	2683 (25)	4366 (24)	0.051
3–8	8933 (28)	51 (25)	126 (26)	173 (18)	248 (27)	3385 (31)	4950 (27)	<0.001
**Injury Severity Score**	17 (16–25)	16 (16–25)	16 (16–25)	17 (16–25)	18 (16–25)	20 (16–25)	17 (16–25)	<0.001
**Revised Trauma Score**	7.84 (5.97–7.84)	6.90 (5.88–7.55)	7.55 (5.97–7.84)	7.84 (6.90–7.84)	7.84 (5.97–7.84)	7.84 (5.97–7.84)	7.84 (5.97–7.84)	<0.001
**TRISS Ps**	0.97 (0.94–0.99)	0.92 (0.79–0.94)	0.98 (0.94–0.99)	0.99 (0.96–0.99)	0.98 (0.94–0.99)	0.94 (0.85–0.98)	0.89 (0.77–0.94)	<0.001
**Actual in-hospital mortality**	5542 (17.3)	17 (8.5)	36 (7.5)	31 (3.2)	56 (6.2)	1579 (14.6)	3823 (21)	<0.001
**Standardized** **mortality ratio**	0.82	0.82	0.81	0.59	0.78	0.94	0.98	0.416

Data are presented as number (percentage) or median (interquartile range Q1–Q3); TRISS, Trauma and Injury Severity Score; Ps, Probability of survival.

**Table 2 jcm-10-01072-t002:** Clinical parameters of patients with traumatic brain injury by age group.

Variables	Overall *n* = 31,953	Neonates/Infants *n* = 201	Preschool Children *n* = 482	Schoolchildren *n* = 964	Adolescents *n* = 910	Adults *n* = 10,823	Older Adults *n* = 18,573	*p*-Value
**Urgent examination**								
Computed tomography	31,208 (98)	191 (95)	464 (96)	936 (97)	893 (98)	10,534 (97)	18,190 (98)	<0.001
**Urgent treatment**								
Blood transfusion within 24 h	4147 (13)	45 (22)	56 (12)	57 (6)	66 (7)	1528 (14)	2395 (13)	<0.001
Transcatheter arterial embolization	106 (0.3)	0	1 (0.2)	0	1 (0.1)	24 (0.2)	80 (0.4)	0.011
Surgery								
Craniotomy	5501 (17)	33 (17)	77 (16)	148 (15)	142 (16)	2236 (22)	2765 (15)	<0.001
Craterization	2155 (7)	14 (7)	21 (4)	20 (2)	46 (5)	558 (5)	1496 (8)	<0.001
**Hospital location**								
Intensive care unit	22,670 (71)	141 (70)	326 (68)	692 (72)	714 (78)	8115 (75)	12,682 (68)	<0.001
General ward	7617 (24)	51 (25)	123 (26)	231 (24)	162 (18)	2109 (19)	4941 (27)	<0.001
**Length of hospital stay, days**	32 (7–97)	32 (6–104)	25 (4–101)	20 (6–93)	32 (9–114)	32 (8–101)	32 (6–94)	<0.001
**Disposition at discharge**								
Home	12,917 (49)	137 (74)	369 (83)	833 (89)	642 (75)	5124 (55)	5811 (39)	<0.001
Hospital transfer	13,002 (49)	35 (19)	71 (16)	95 (10)	207 (24)	4028 (44)	8566 (58)	<0.001

Data are presented as number (percentage) or median (interquartile range Q1–Q3).

**Table 3 jcm-10-01072-t003:** Multivariate logistic regression analysis of in-hospital mortality among patients with severe traumatic brain injury.

	Overall (*n* = 31,953)			Patients with a GCS Score of 3–8 (*n* = 8933)		
	OR	(95 % CI)	*p*-Value	OR	(95 % CI)	*p*-Value
**Age in years**						
0 year	0.40	(0.21–0.76)	0.006	0.50	(0.23–1.10)	0.086
1–5 years	0.37	(0.24–0.57)	<0.000	0.62	(0.38–1.01)	0.053
6–12 years	0.23	(0.15–0.36)	<0.000	0.39	(0.24–0.64)	<0.000
13–17 years	0.38	(0.27–0.53)	<0.000	0.49	(0.33–0.73)	<0.000
18–64 years	0.69	(0.63–0.76)	<0.000	0.86	(0.76–0.96)	0.008
≥65 years	1.00	−		1.00	−	
**Year of hospital admission**						
2009	1.64	(1.35–1.98)	<0.000	1.28	(0.99–1.64)	0.055
2010	1.52	(1.27–1.82)	<0.000	1.38	(1.09–1.75)	0.007
2011	1.29	(1.08–1.53)	0.005	1.23	(0.98–1.55)	0.069
2012	1.40	(1.19–1.65)	<0.000	1.27	(1.02–1.57)	0.028
2013	1.23	(1.05–1.44)	0.012	1.13	(0.92–1.39)	0.254
2014	1.11	(0.94–1.26)	0.213	1.12	(0.91–1.38)	0.283
2015	1.07	(0.92–1.26)	0.389	1.93	(0.84–1.26)	0.805
2016	1.05	(0.89–1.24)	0.583	1.01	(0.81–1.24)	0.959
2017	1.04	(0.88–1.23)	0.637	1.07	(0.86–1.33)	0.551
2018	1.00	−		1.00	−	
**Transport type**						
Transportation from the scene	1.17	(0.98–1.40)	0.091	1.25	(0.94–1.66)	0.131
Transportation from another hospital	0.81	(0.66–0.99)	0.048	1.00	(0.73–1.37)	0.984
**Injury mechanism of blunt trauma**						
Traffic accident						
Motor vehicle	0.65	(0.51–0.83)	<0.000	0.57	(0.42–0.77)	<0.000
Bike	0.65	(0.52–0.80)	<0.000	0.67	(0.52–0.86)	0.002
Bicycle	0.89	(0.76–1.05)	0.184	0.91	(0.74–1.11)	0.347
Pedestrian	0.98	(0.83–1.16)	0.831	0.93	(0.76–1.14)	0.466
Fall from a height	1.36	(1.13–1.64)	0.001	1.47	(1.16–1.87)	0.002
Fall from stairs	0.92	(0.79–1.06)	0.257	0.87	(0.73–1.05)	0.139
Fall at a same level	1.12	(0.97–1.28)	0.113	0.90	(0.75–1.06)	0.204
Others	1.00	−		1.00	−	
**Glasgow Coma Scale score on hospital arrival**						
14, 15	0.19	(0.16–0.22)	<0.000			
9–13	0.51	(0.45–0.57)	<0.000			
3–8	1.00	−				
**TRISS Ps**	0.01	(0.01–0.02)	<0.000	0.02	(0.01–0.02)	<0.000
**Urgent treatment**						
Blood transfusion within 24 h	2.13	(1.91–2.37)	<0.000	1.77	(1.55–2.03)	<0.000
Transcatheter arterial embolization	0.37	(0.19–0.72)	0.003	0.28	(0.12–0.65)	0.003
Surgery						
Craniotomy	0.50	(0.45–0.55)	<0.000	0.31	(0.27–0.35)	<0.000
Craterization	0.84	(0.74–0.97)	0.014	0.69	(0.59–0.81)	<0.000

OR, odds ratio; CI, confidence interval; TRISS, Trauma and Injury Severity Score; Ps, Probability of survival.

## Data Availability

The approving authority for data access was the Japanese Association for the Surgery of Trauma (Trauma Registry Committee). The date presented in this study are available on request from the corresponding author to Trauma Registry Committee.

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
