# Peer review of "Age- and Severity-Related In-Hospital Mortality Trends and Risks of Severe Traumatic Brain Injury in Japan: A Nationwide 10-Year Retrospective Study"

_jcm, 2021, doi:10.3390/jcm10051072_

Round 1

Reviewer 1 Report

In this retrospective nationwide study of traumatic brain injury (TBI) from Japan, the authors aimed to evaluate the age, severity-related in-hospital mortality trends, and mortality risks of patients with severe TBI (sTBI) from the year 2009 to 2018. Data were obtained from their extensive Japan Trauma Data Bank dataset (JTDB). The authors mentioned that the general purpose of this study to establish effective injury prevention measures. Overall, the authors declare; (1) a significant age-related difference in characteristics, mortality trend, and mortality risk in patients with sTBI, (2) in-hospital mortality trends of all patients with severe TBI significantly decreased along the reviewed years, (3) in patients with severe TBI but ≥ 65 years old with a fall from a height, and need for an urgent blood transfusion, their mortality risk was higher, and (4) patients with ≤ 5 or ≥ 65 years-old their mortality risk did not decrease significantly. The authors concluded that appropriate neurosurgical care may reduce mortality in this high-risk patient population.

The study is well-written and designed, clearly detailing inclusions and exclusions criteria. The subgroup stratification is commonly used by authors in this research field. The title is adequate to the content of the article. The size of the cohorts described is significant to reach general extrapolations. The statistical model and analysis of the clinical data seem appropriate. The discussion points are properly linked to the results presented, and the results are well-presented to reach reasonable conclusions. Moreover, the authors properly mentioned several limitations, as well future directions that need to be considered. Overall, this study seems to be carried by authors with significant experience in this field, and this work represents a reasonable extension from previous trauma studies. However, some minor concerns need to be addressed before this article can move forward.

1. In this study, the authors aimed to evaluate in-hospital mortality trends and risks of patients with sTBI to establish effective injury prevention measures. Although the authors mentioned that “appropriate neurosurgical care” may reduce mortality in the sTBI patient population, it is not clear (based on the data presented) if patients who underwent neurosurgical treatments had better outcomes than patients who only were exposed to medical treatments. Might be future studies specifically comparing sTBI outcomes (studying groups with similar brain lesions) could address such a statement.

2. Although extraneuronal trauma was part of the exclusion criteria, based on their previous work with pediatric general trauma patients (Toida, C. et al 2020; Journal of Clinical Medicine), there are common mortality risk factors between TBI and trauma populations (as an example, urgent blood transfusion). In that regard, are the authors considering other unaccounted extra-neurological mechanisms (coagulopathies, ionic problems, etc.) that could increase the mortality in the sTBI groups? You may briefly discuss this point.

3. The classification of surgical treatments in this work is only based on craniotomy and catheterization, which could be somewhat oversimplistic to account for the variety of potential outcomes from the ample spectrum of neurosurgical procedures. As an example, different neurosurgical pathologies, such as acute epidural hematoma, acute subdural hematoma, chronic subdural hematomas, intracranial hemorrhage, etc. have different relevance considering specific demographic factors, as well as baseline medical conditions. Maybe the authors could consider addressing this in future studies.

4. The authors mentioned the Glasgow come scale (GCS) as a unique tool for the entire population. However, it is implicit that the authors need to normalize their results with other versions such as the modified GCS for infants and children. The clarification of this point is important, as original studies suggested that sTBI outcomes are different across different age groups (Lieh-Lai, MW et al. 1992; The Journal of Pediatrics).

5. From the study setting and population, is not clear if mortality due to intra-hospital complications (infection, hypoxia, TBI-related arrhythmias, and cardiovascular arrest, among other) and medical therapeutic strategies (hyperventilation, neuroprotective agents, induced coma, etc.,) were recorded and analyzed on their database.

6. Based on Table 2, it seems adults and older adults have a greater percentage of hospital transfers, particularly compared to groups at younger ages. Is it related to each hospital, local, or national health policies? A short comment will suffice.

7. In figure 2, the percentage per year-in mortality for patients more than 65 years-old was significantly decreased over time (26% to 17.7%). However, the incidence of sTBI in the patient population doubled in 10 years (212 to 429). Is this related to the general aging phenomena across the Japanese population? If it is so, could be possible that an older subgroup (more than 75-year-old) need to be considered in future studies? (see Thompson, H.J et al 2006; Journal of American Geriatric Society).

8. Although traffic accidents by motor vehicles or bikes were significantly associated with lower mortality risk, the data support a greater incidence of traffic accidents among bicycle riders and pedestrians. Thus, including traffic policies and increased awareness could reduce the impact on mortality as well. A short comment to increase awareness of these facts could be considered.

9. Some studies have noted the relative value of GCS as a predictor of neurological outcomes in the diagnosis, surgical management, and posterior prognosis. This has been established comparing GSC to other scales, such as the full outline of unresponsiveness scale (FOUR) (Jalali, R. et al. 2014; Critical Care Research and Practice & Eelco, W. et al; Neurologic Critical Care). Please discuss briefly.

Minor remarks:

10. On page 4 of 12, line 202; “…. patient with a GCS score of 3–8 was 17.3% and 47.9%, ...” or “…. patient with a GCS score of 3–8 was 17.3% to 9%? Please revise.

11. On page 4 of 12, line 205 for the sentence; “…. the mortality trend of severe TBI patients aged less than 5 years...” do the authors mean “…. the mortality trend of TBI patients severe with less than 5 years-old….”? The word choice is confusing. Please revise if with “aged” (the process of becoming older), the authors meant “years-old” (the age of something or someone, in terms of years) Moreover, in line 211 “with a GCS score of 3–8 aged = 5 years” is not clear.

12. On page 4 of 12, line 205; in the written segment: “… In particular, improving the mortality risk of severe TBI patients with a GCS score of 3–8 was high may be more effective in decreasing the mortality because the mortality risk of patients with a GCS score of 3–8 aged = with ≤ 5 years or ≥ 65 years was higher than that of other age groups” is long and unclear. Please revise using shorter and clear sentences.

13. On page 4, line 248 the phrase; “…’ It is still under better debate, direct transfer with neurosurgical facilities from the scene…” is not clear. Its means " direct transfer from (or to) neurosurgical facilities from the scene..."? Please revise.

14. In the section of the author's contribution, the authors mentioned in the tile “Patents”? Please revise.

In sum, the current study represents a potentially useful tool to evaluate the incidence and evolution of the sTBI, as well as potential risk factors, allocation of resources, and preventive measures in this particular patient population.

Author Response

Reviewer’s comments (in blue) and Answers (in black)

We wish to express our deep appreciation for the valuable comments of the Reviewer regarding our manuscript. We believe that our manuscript has greatly benefited from these comments.

Reviewer: 1

  1. In this study, the authors aimed to evaluate in-hospital mortality trends and risks of patients with sTBI to establish effective injury prevention measures. Although the authors mentioned that “appropriate neurosurgical care” may reduce mortality in the sTBI patient population, it is not clear (based on the data presented) if patients who underwent neurosurgical treatments had better outcomes than patients who only were exposed to medical treatments. Might be future studies specifically comparing sTBI outcomes (studying groups with similar brain lesions) could address such a statement.

Response: We thank the Reviewer for this valuable comment. We agree with this comment and suggestion. Accordingly, we have revised the Conclusion section as follows:

“Patients with severe TBI, a GCS score between 3-8, and those who need an urgent blood transfusion and surgical intervention might have better outcomes if they receive specific treatment strategies.” (lines 315–317)

  1. Although extraneuronal trauma was part of the exclusion criteria, based on their previous work with pediatric general trauma patients (Toida, C. et al 2020; Journal of Clinical Medicine), there are common mortality risk factors between TBI and trauma populations (as an example, urgent blood transfusion). In that regard, are the authors considering other unaccounted extra-neurological mechanisms (coagulopathies, ionic problems, etc.) that could increase the mortality in the sTBI groups? You may briefly discuss this point.

Response: We thank the Reviewer for this comment. We agree that other extra-neurological mechanisms may affect mortality in TBI patients. However, the JTDB registry did not include detailed blood examination data. Accordingly, we have included this point in the limitation section of the manuscript as follows:

“Therefore, in the future, we should analyze the specific mortality risk of patients with severe TBI considering other unaccounted extra-neurological mechanisms, such as coagulopathies or ionic problems, as well as appropriate strategies to reduce severe TBI mortality of at-risk patients.” (lines 301–304)

  1. The classification of surgical treatments in this work is only based on craniotomy and catheterization, which could be somewhat oversimplistic to account for the variety of potential outcomes from the ample spectrum of neurosurgical procedures. As an example, different neurosurgical pathologies, such as acute epidural hematoma, acute subdural hematoma, chronic subdural hematomas, intracranial hemorrhage, etc. have different relevance considering specific demographic factors, as well as baseline medical conditions. Maybe the authors could consider addressing this in future studies.

Response: We thank the Reviewer for this comment. We agree that different neurosurgical pathologies have different relevance, which was an aspect that we could not evaluate in this study. Accordingly, we have included this point in the limitation section of the manuscript as follows:

“Moreover, different neurosurgical pathologies—such as acute epidural and subdural hematomas, chronic subdural hematoma, and intracranial hemorrhage—have distinct relevance depending on specific demographic factors and baseline medical conditions. Therefore, in the future, we should analyze the specific mortality risk of patients with severe TBI considering other unaccounted extra-neurological mechanisms, such as coagulopathies or ionic problems, as well as appropriate strategies to reduce severe TBI mortality of at-risk patients.” (lines 298–304)

  1. The authors mentioned the Glasgow come scale (GCS) as a unique tool for the entire population. However, it is implicit that the authors need to normalize their results with other versions such as the modified GCS for infants and children. The clarification of this point is important, as original studies suggested that sTBI outcomes are different across different age groups (Lieh-Lai, MW et al. 1992; The Journal of Pediatrics).

Response: We thank the Reviewer for this valuable comment. We agree that the additional information on the GCS would be important for the understanding of the manuscript. Accordingly, we have revised the Discussion section to include this clarification as follows:

“Previous studies indicated the limitations of using the GCS as a predictor of neurological outcomes for diagnosis, surgical management, and posterior prognosis, especially in younger patients and those with hypoxic injury [22,23]. Although we applied the GCS universally in this study, future studies may have to normalize the results in relation to other scales, such as the modified GCS for young children or the full outline of unresponsiveness scale.” (lines 267–272)

  1. From the study setting and population, is not clear if mortality due to intra-hospital complications (infection, hypoxia, TBI-related arrhythmias, and cardiovascular arrest, among other) and medical therapeutic strategies (hyperventilation, neuroprotective agents, induced coma, etc.,) were recorded and analyzed on their database.

Response: We thank the Reviewer for this comment. Although intra-hospital complications and medical therapeutic strategies can affect mortality, the JTDB registry does not include information regarding these aspects. Thus, we could not analyze the relationship between mortality and intra-hospital complications and therapeutic strategies. We have revised the limitation section of the manuscript to include this aspect as follows:

“Finally, we could not assess the differences in mortality owing to intrahospital complications and medical therapeutic strategies among the facilities.” (lines 304–306)

  1. Based on Table 2, it seems adults and older adults have a greater percentage of hospital transfers, particularly compared to groups at younger ages. Is it related to each hospital, local, or national health policies? A short comment will suffice.

Response: We thank the Reviewer for this comment. Although we observed a difference in the proportion of inter-hospital transfers according to age group, local and national health policy regarding EMS do not differ among them. We have revised the Discussion section of the manuscript to address this as follows:

“Whether a direct transfer from the scene to a neurosurgical facility or secondary transfer from a general hospital to a neurosurgical facility is the most appropriate course of action remains unclear. Moreover, we observed that the proportion of interhospital transfer differed according to age group; thus, a future analysis must assess the effect of different transfers on mortality.” (lines 263–267)

  1. In figure 2, the percentage per year-in mortality for patients more than 65 years-old was significantly decreased over time (26% to 17.7%). However, the incidence of sTBI in the patient population doubled in 10 years (212 to 429). Is this related to the general aging phenomena across the Japanese population? If it is so, could be possible that an older subgroup (more than 75-year-old) need to be considered in future studies? (see Thompson, H.J et al 2006; Journal of American GeriatricSociety).

Response: We thank the Reviewer for this insightful suggestion. In our study, we observed that the proportion of severe TBI in older adults increased from 47% in 2009 to 64% in 2018. We believe that this increase can be mainly attributed to the aging of the Japanese population. In addition, we agree that future studies should include a subgroup analysis to investigate this tendency. Accordingly, we have revised the limitation section of our study to address this as follows:

“Our results indicate that the aging of the Japanese population probably affected the proportion of older patients with severe TBI, which increased from 47% in 2009 to 64% in 2018. A previous report showed that patients aged ≥75 years had the highest TBI-related mortality among all age groups [23]. Thus, future studies may consider conducting sub-analyses within the aged group.” (lines 279–283)

  1. Although traffic accidents by motor vehicles or bikes were significantly associated with lower mortality risk, the data support a greater incidence of traffic accidents among bicycle riders and pedestrians. Thus, including traffic policies and increased awareness could reduce the impact on mortality as well. A short comment to increase awareness of these facts could be considered.

Response: We thank the Reviewer for this comment. As requested, we have revised the Discussion section to address this aspect as follows:

“Although traffic accidents involving motor vehicles or bikes were significantly associated with a lower mortality risk, we observed a high frequency of traffic accidents involving bicycle riders and pedestrians. Thus, traffic policies focused on these modes of transportation may positively affect mortality rates.” (lines 240–243)

  1. Some studies have noted the relative value of GCS as a predictor of neurological outcomes in the diagnosis, surgical management, and posterior prognosis. This has been established comparing GSC to other scales, such as the full outline of unresponsiveness scale (FOUR) (Jalali, R. et al. 2014; Critical Care Research and Practice & Eelco, W. et al; Neurologic Critical Care). Please discuss briefly.

Response: We thank the Reviewer for this valuable comment. We agree that the additional information on the GCS would be important for the understanding of the manuscript. Accordingly, we have revised the manuscript to address this as follows:

“Previous studies indicated the limitations of using the GCS as a predictor of neurological outcomes for diagnosis, surgical management, and posterior prognosis, especially in younger patients and those with hypoxic injury [22,23]. Although we applied the GCS universally in this study, future studies may have to normalize the results in relation to other scales, such as the modified GCS for young children or the full outline of unresponsiveness scale.” (lines 267–272)

Minor remarks:

  1. On page 4 of 12, line 202; “…. patient with a GCS score of 3–8 was 17.3% and 47.9%, ...” or “…. patient with a GCS score of 3–8 was 17.3% to 9%? Please revise.

Response: We thank the Reviewer for this comment. This error has been corrected as follows:

“ […] we observed in-hospital mortality rates of 46.4% and 49.6% in all patients with severe TBI and those with a GCS score between 3–8, respectively […]” (lines 214–216)

  1. On page 4 of 12, line 205 for the sentence; “…. The mortality trend of severe TBI patients aged less than 5 years...” do the authors mean “…. the mortality trend of TBI patients severe with less than 5 years-old….”? The word choice is confusing. Please revise if with “aged” (the process of becoming older), the authors meant “years-old” (the age of something or someone, in terms of years) Moreover, in line 211 “with a GCS score of 3–8 aged = 5 years” is not clear.

Response: We thank the Reviewer for this comment. We have revised the sentences to improve their clarity as follows:

“The mortality trend of patients with severe TBI that are younger than 5 years old […]” (lines 217–218)

“Regarding why young patients with a low GCS score had a higher mortality risk […]” (lines 223–224)

  1. On page 4 of 12, line 205; in the written segment: “… In particular, improving the mortality risk of severe TBI patients with a GCS score of 3–8 was high may be more effective in decreasing the mortality because the mortality risk of patients with a GCS score of 3–8 aged = with ≤ 5 years or ≥ 65 years was higher than that of other age groups” is long and unclear. Please revise using shorter and clear sentences.

Response: We thank the Reviewer for this comment. As requested, we have revised this portion as follows:

“Ensuring that patients with a low GCS score—especially those that are also younger than 5 years old or older than 65 years old—receive specific and appropriate medical treatment considering their risk factors may effectively decrease the mortality rate of patients with severe TBI.” (lines 220–223)

  1. On page 4, line 248 the phrase; “…’ It is still under better debate, direct transfer with neurosurgical facilities from the scene…” is not clear. Its means " direct transfer from (or to) neurosurgical facilities from the scene..."? Please revise.

Response: We thank the Reviewer for this comment. As requested, we have revised this sentence as follows:

“Whether a direct transfer from the scene to a neurosurgical facility or secondary transfer from a general hospital to a neurosurgical facility is the most appropriate course of action remains unclear.” (lines 263–265)

  1. In the section of the author's contribution, the authors mentioned in the tile “Patents”? Please revise.

Response: We thank the Reviewer for this comment. Because there are no patents, we have deleted the heading.

  1. In sum, the current study represents a potentially useful tool to evaluate the incidence and evolution of the sTBI, as well as potential risk factors, allocation of resources, and preventive measures in this particular patient population.

Response: We thank the Reviewer for the opportunity to strengthen our manuscript with their valuable comments and suggestions. We have worked diligently to incorporate the feedback into the revised manuscript and hope that the manuscript now meets all requirements for publication.

Reviewer 2 Report

The authors describe the trend in the in-hospital mortality rate related to age and severity of injury in the past 10 years in Japan. The topic is really interesting and the data collected worthy to be published.

I have a few comments on how to present and discuss the results obtained.

I am not satisfied with the introduction. I cannot understand if the application of the data presented is designed to improve the primary prevention of brain trauma to the groups with worse prognosis (e.g. new regulation to protect the juvenile population from abuses) or secondary prevention to guide more aggressive and quick treatments to patients with specific risk factors in age and/or severity of injury. If this study is designed to enhance the prevention of secondary injury, a paragraph to explain current treatments and clinical challenges in the management of brain trauma can be useful to better frame the aims of the studies. Some clinical studies failed to improve patients’ prognoses using different modalities of treatment (decompressive craniectomy, hypothermia) (1, 2), would an analysis of patients’ risk factors guide clinicians to a more “tailored” approach?

Lines 43-44: the status of the population in Japan is really important in the analysis of the data shown, thus this sentence needs a reference.

Lines 40-42: as I stated above, I cannot understand if “injury prevention measures” are referring to primary and secondary prevention. The reference used did not help as it is too broad on the topic. The introduction can be more clearly phrased and more specific references added.

LINES 70-72: The ISS score system used to assess the severity of the injury has some limitations, especially related to the probability of mortality (3). Can the ISS score limitations addressed and/or mentioned?

Lines 217-219: please see the points above related to the use of this data to help to guide the treatment.

Lines 204: I am interested in the decrease in the mortality rate in the general TBI population, however, the authors focus primarily on the absence of this decrease in the population less than 5 years old. Can the authors hypothesize why there was this decrease? Is it limited to the developed countries? Without getting into too many details, a better analysis of this relevant result can benefit the paper.

207: there is a typo, Japan should be in capital letters.

277: The conclusions can be better stated if the aims of the study are framed better.

  1. Cooper DJ, Rosenfeld JV, Murray L, Arabi YM, Davies AR, D'Urso P, et al. Decompressive craniectomy in diffuse traumatic brain injury. N Engl J Med. 2011;364(16):1493-502.
  2. Andrews PJ, Sinclair HL, Rodriguez A, Harris BA, Battison CG, Rhodes JK, et al. Hypothermia for Intracranial Hypertension after Traumatic Brain Injury. N Engl J Med. 2015;373(25):2403-12.
  3. Moore EE, Feliciano DV, Mattox KL. Trauma, Eighth Edition: McGraw-Hill Education; 2017.

Author Response

Reviewer’s comments (in blue) and Answers (in black)

We wish to express our deep appreciation for the valuable comments of the Reviewer regarding our manuscript. We believe that our manuscript has greatly benefited from these comments.

Reviewer: 2

  1. I am not satisfied with the introduction. I cannot understand if the application of the data presented is designed to improve the primary prevention of brain trauma to the groups with worse prognosis (e.g. new regulation to protect the juvenile population from abuses) or secondary prevention to guide more aggressive and quick treatments to patients with specific risk factors in age and/or severity of injury. If this study is designed to enhance the prevention of secondary injury, a paragraph to explain current treatments and clinical challenges in the management of brain trauma can be useful to better frame the aims of the studies. Some clinical studies failed to improve patients’ prognoses using different modalities of treatment (decompressive craniectomy, hypothermia) (1, 2), would an analysis of patients’ risk factors guide clinicians to a more “tailored” approach?

Response: We thank the Reviewer for this comment. We agree with the suggestion. While we focused on the assessment of risk factors related to primary and secondary prevention, the limitations mentioned in the comment are important to address. Accordingly, we have revised several sections of the manuscript to address the Reviewer’s concerns as follows:

“Therefore, injury surveillance is essential for identifying age- and severity-related injury characteristics and monitoring in-hospital mortality trends over time, as well as to assess risk factors related to primary and secondary injury prevention, to estimate the effectiveness of injury prevention measures [13,14].” (lines 39–43)

“Thus, this study aimed to evaluate age- and severity-related in-hospital mortality trends and risk factors associated with mortality in patients with severe TBI using nationwide data comprising a 10-year period between 2009–2018. We aimed to address the aforementioned research gaps as well as to provide information that could help developing and evaluating effective injury prevention measures and specific therapeutic strategies.” (lines 50–55)

“Previous studies failed to prove that the outcomes of patients improved with the use of simple risk factor assessments [27, 28]. Accordingly, although this retrospective study assessed the mortality risk of patients with severe TBI, our results regarding mortality risk cannot be used to directly guide clinicians toward optimal TBI prevention and treatment strategies. Moreover, different neurosurgical pathologies—such as acute epidural and subdural hematomas, chronic subdural hematoma, and intracranial hemorrhage—have distinct relevance depending on specific demographic factors and baseline medical conditions. Therefore, in the future, we should analyze the specific mortality risk of patients with severe TBI considering other unaccounted extra-neurological mechanisms, such as coagulopathies or ionic problems, as well as appropriate strategies to reduce severe TBI mortality of at-risk patients.” (lines 294–304)

“Future nationwide studies with subclass analyses should be conducted to provide information for developing these strategies as well as prevention measures to improve the outcomes of patients with severe TBI.” (lines 317–320)

  1. Lines 43-44: the status of the population in Japan is really important in the analysis of the data shown, thus this sentence needs a reference.

Response: We thank the Reviewer for this comment. As requested, we cited references 3 and 12 in the sentence.

  1. Lines 40-42: as I stated above, I cannot understand if “injury prevention measures” are referring to primary and secondary prevention. The reference used did not help as it is too broad on the topic. The introduction can be more clearly phrased and more specific references added.

Response: We thank the Reviewer for this valuable comment. As requested, we have revised the Introduction section as follows:

“Therefore, injury surveillance is essential for identifying age- and severity-related injury characteristics and monitoring in-hospital mortality trends over time, as well as to assess risk factors related to primary and secondary injury prevention, to estimate the effectiveness of injury prevention measures [13,14].” (lines 39–43)

“Thus, this study aimed to evaluate age- and severity-related in-hospital mortality trends and risk factors associated with mortality in patients with severe TBI using nationwide data comprising a 10-year period between 2009–2018. We aimed to address the aforementioned research gaps as well as to provide information that could help developing and evaluating effective injury prevention measures and specific therapeutic strategies.” (lines 50–55)

  1. LINES 70-72: The ISS score system used to assess the severity of the injury has some limitations, especially related to the probability of mortality (3). Can the ISS score limitations addressed and/or mentioned?

Response: We thank the reviewer for this comment. As requested, we have revised the limitation section of the manuscript to address the limitations of the ISS score system as follows:

“Moreover, the ISS score system used to assess injury severity may have some limitations, especially related to mortalityprobability or scoring system validity [25,26].” (lines 283–285)

  1. Lines 217-219: please see the points above related to the use of this data to help to guide the treatment.

Response: Thank you for your valuable comment. We have added the following sentence as the limitation of this study.

Response: We thank the Reviewer for this valuable comment. As requested, we have revised the limitation section of the manuscript to address these points as follows:

“Previous studies failed to prove that the outcomes of patients improved with the use of simple risk factor assessments [27, 28]. Accordingly, although this retrospective study assessed the mortality risk of patients with severe TBI, our results regarding mortality risk cannot be used to directly guide clinicians toward optimal TBI prevention and treatment strategies. Moreover, different neurosurgical pathologies—such as acute epidural and subdural hematomas, chronic subdural hematoma, and intracranial hemorrhage—have distinct relevance depending on specific demographic factors and baseline medical conditions. Therefore, in the future, we should analyze the specific mortality risk of patients with severe TBI considering other unaccounted extra-neurological mechanisms, such as coagulopathies or ionic problems, as well as appropriate strategies to reduce severe TBI mortality of at-risk patients.” (lines 294–304)

  1. Lines 204: I am interested in the decrease in the mortality rate in the general TBI population, however, the authors focus primarily on the absence of this decrease in the population less than 5 years old. Can the authors hypothesize why there was this decrease? Is it limited to the developed countries? Without getting into too many details, a better analysis of this relevant result can benefit the paper.

Response: We thank the Reviewer for this valuable suggestion. We agree that additional analysis and discussion regarding the mortality trend of patients younger than 5 years old would be beneficial to the manuscript. Regrettably, however, we could not find relevant studies and data to support further discussion of these points.

  1. 207: there is a typo, Japan should be in capital letters.

Response: We thank the Reviewer for this comment. The typing error has been corrected.

  1. 277: The conclusions can be better stated if the aims of the study are framed better.

Response: We thank the reviewer for this suggestion. We have revised the Conclusion section as follows:

“Future nationwide studies with subclass analyses should be conducted to provide information for developing these strategies as well as prevention measures to improve the outcomes of patients with severe TBI.” (lines 317–320)
